# Distribution and Implications of Haloarchaeal Plasmids Disseminated in Self-Encoded Plasmid Vesicles

**DOI:** 10.3390/microorganisms12010005

**Published:** 2023-12-19

**Authors:** Dominik Lücking, Tomás Alarcón-Schumacher, Susanne Erdmann

**Affiliations:** Max-Planck Institute for Marine Microbiology, Celsiusstrasse 1, 28359 Bremen, Germany

**Keywords:** archaea, metagenomics, mobile genetic elements, plasmids, haloarchaea

## Abstract

Even though viruses and plasmids are both drivers of horizontal gene transfer, they differ fundamentally in their mode of transfer. Virus genomes are enclosed in virus capsids and are not dependent on cell-to-cell contacts for their dissemination. In contrast, the transfer of plasmids most often requires physical contact between cells. However, plasmid pR1SE of *Halorubrum lacusprofundi* is disseminated between cells, independent of cell-cell contacts, in specialized membrane vesicles that contain plasmid proteins. In this study, we searched for pR1SE-like elements in public databases and a metagenomics dataset from Australian salt lakes and identified 40 additional pR1SE-like elements in hypersaline environments worldwide. Herein, these elements are named apHPVs (archaeal plasmids of haloarchaea potentially transferred in plasmid vesicles). They share two sets of closely related proteins with conserved synteny, strongly indicating an organization into different functional clusters. We find that apHPVs, besides transferring themselves, have the potential to transfer large fragments of DNA between host cells, including virus defense systems. Most interestingly, apHPVs likely play an important role in the evolution of viruses and plasmids in haloarchaea, as they appear to recombine with both of them. This further supports the idea that plasmids and viruses are not distinct but closely related mobile genetic elements.

## 1. Introduction

The genomes of bacteria and archaea are subjected to constant change. Vertical gene transfer and mutations allow the continuous adaptation of organisms to their environment. Horizontal gene transfer (HGT) events introduce additional flexibility by transferring larger segments of genetic material between organisms, sometimes even across domain borders [1,2]. These events are most often driven by mobile genetic elements (MGEs). A plethora of MGEs, such as viruses, plasmids, transposons, retrotransposons, and gene transfer agents (GTAs), shape microbial communities at a fundamental level. Traditionally, these MGEs have been described as distinct entities. However, the discovery of elements that break traditional definitions led to the view of a more continuous sequence space [3].

In 2017, Erdmann et al. isolated and characterized the ~50 kbp MGE, plasmid pR1SE, of the halophilic archaeon (haloarchaeon) *Halorubrum lacusporufundi* R1S1 [4]. Plasmids are usually transferred between cells as unprotected DNA or by conjugation, requiring cell-to-cell contact. However, pR1SE is transferred by a mechanism that is more similar to the dissemination of viruses. *Hrr. lacusprofundi* produces extracellular vesicles (EVs), as do other haloarchaea [5]. Plasmid pR1SE, if present, is enclosed into EV-like structures, which are then taken up by plasmid-free strains, thus transferring pR1SE without direct cell-to-cell contact. While transfer of plasmids in EVs was observed previously in archaea, these plasmid-containing vesicles were found to contain exclusively host proteins [6]. However, pR1SE-containing EVs were shown to include pR1SE-encoded proteins that are proposed to be involved in the formation of these vesicles. We consider them plasmid vesicles (PVs), as they can be used to infect a plasmid-free strain that subsequently produces PVs without cell lysis, thus mimicking the life cycle of a chronic viral infection [7]. Plasmid proteins included in PVs were not found to be homologous to any known virus capsid protein and also did not show similarities to proteins involved in conjugation [4]. The ambivalent set of characteristics of pR1SE makes it an intriguing case for the evolutionary trajectory of archaeal plasmids and viruses.

Plasmid pR1SE was also found to integrate into the host genome and incorporate and transfer large fragments of host genomic DNA [4], indicating that it could play a major role in the remarkable HGT described for haloarchaeal host organisms [8]. Haloarchaea are known to exhibit secondary chromosomes and megaplasmids [9], and it has been hypothesized that pR1SE could be involved in the emergence of these additional replicons [4]. However, only one element with similarities to pR1SE was detected in public databases by BlastP searches upon the discovery of pR1SE in 2017. Therefore, in this study, we used a hidden Markov model (HMM)-based approach to discover pR1SE-like plasmids. HMM profiles contain the evolutionary history of proteins by calculating transition probabilities from one state of a sequence to a mutated iteration and have been shown to improve the sensitivity of searches. By analyzing metagenomes from Australian salt lakes and searching public databases, we identified a set of pR1SE-like elements. This enabled the description of the general genomic organization of the plasmid, leading to a better understanding of the functional roles of conserved proteins. We describe the core proteins in detail and offer hypotheses on the function of two distinct core gene clusters as well as the ecological and evolutionary impact of PV-mediated HGT.

## 2. Materials and Methods

### 2.1. Sampling of Australian Salt Lakes, DNA Extraction, and Sequencing

From December 2018 to January 2019, the sediment salt crust of eleven hypersaline lakes was sampled with permission from the Department for Environment and Water, South Australia (Permission number: U26817-1) and the Department of Environment, Land, and Water Planning, Victoria (Permission number: 1008945). Subsamples of 1 g dry-weight sediment for each lake were used for DNA extraction using the FastDNA™ SPIN Kit for Soils. DNA sequencing libraries were prepared (FS DNA Library, NEBNext^®^ Ultra™) and sequenced (2 × 250 bp paired end, Illumina HiSeq2500, Rapid Mode) at the Max-Planck Genome-Centre Cologne (Germany). The data have previously been described in Alarcón-Schumacher et al. [10], and raw data are available at ENA-EMBL under project number PRJEB61734. Reads were trimmed with Cutadapt (v4.6) [11], removing low-quality, short, or unpaired reads (parameters: -q 30 -m 30). The remaining reads were assembled with metaSPAdes v3.13.1 [12].

### 2.2. Creation of pR1SE Protein Clusters

Hidden-Markov-Model (HMM) profiles were generated for six open reading frames of pR1SE (ORF6, ORF8, ORF17, ORF21, ORF23, and ORF24) as follows, also described in Appendix A. Input proteins (Appendix A) were iteratively compared by BLASTp against the nr database of non-redundant proteins (accessed on 18 April 2023) using diamond blastp [13]. Proteins with a bitscore above 50 and an evalue below 10^−5^ were downloaded. Sequences were aligned using mafft [14] (–localpair –reorder), and upon manual inspection of the alignment, proteins that introduced larger gaps in the alignment or extended the alignment on either side were removed. A coverage threshold was established (Appendix A). Sequences that were removed in this step were added to an ‘accession blacklist’ and were not considered in further alignments. The following steps were carried out iteratively until no further proteins could be added to an ORF cluster: First, multiple sequence alignments were generated using mafft [14] with the –localpair –reorder flags. The resulting alignment was visually inspected. Proteins that extended over the edges of the original alignment or introduced large gaps within the alignment were removed and blacklisted, in order to avoid the artificial inflation of the search space. Then, HMM profiles of the alignments were generated using hmmbuild, part of the hmmer tool package (v3.3.2) [15]. The resulting profiles were then searched against nr using hmmsearch from the same package. Finally, hits above an ORF-dependent bitscore and coverage threshold (Appendix A) were retrieved and added to the ORF cluster. After three (ORF6, ORF17, ORF21, ORF23, ORF24) or five (ORF8) iterations, no further proteins could be retrieved, and a final HMM profile was created.

### 2.3. Detection of pR1SE-like Elements (apHPVs) in Public Databases and Metagenomes from Australian Salt Lakes

Using the pR1SE protein cluster HMM profiles, the Australian salt lake contigs were searched for pR1SE relatives (hmmsearch, score according to Appendix A). Contigs longer than 10,000 bp that showed at least three hits were selected for detailed analysis (N = 18). Similarly, all genomes associated with the proteins retrieved from the nr database when generating the HMM profiles were downloaded from NCBI using the Rentrez library (David Winter, https://CRAN.R-project.org/package=rentrez, accessed on 3 August 2023) and searched using the HMM profiles (hmmsearch, –score 50). Again, contigs longer than 10,000 bp that showed at least three hits were considered for detailed analysis (N = 63). Similarly, the IMG/VR database (v4.1 release, December 2022) [16] was searched for potential candidates. Contigs with at least five of six core genes were considered “complete” pR1SE-like elements. In total, 40 pR1SE-like elements, in addition to the original pR1SE plasmid, were retrieved and, from now on, termed apHPVs (see Results Section 3.2) (Appendix A).

Host information on NCBI apHPVs was retrieved from NCBI. Host prediction for the apHPVs from Australian salt lakes was carried out using iPhoP (https://bitbucket.org/srouxjgi/iphop/src/main/, accessed on 12 September 2023 [17]) in predict mode. The circularity information of the contigs was either directly retrieved from NCBI or determined manually by searching for terminal repeats longer than 30 bp at both ends of the contig. Direct, palindromic, or interspersed direct repeats were detected using the ‘Find repeats’ function in Geneious (Build 2022-08-18). Repeats > 25 bp with a maximum of 1 mismatch were considered positive identifications.

### 2.4. Identification of the apHPV Core Genes

In order to identify the core genes of apHPVs, we selected ten genes upstream of ORF6, all genes between and including ORF6 and ORF24, and the 10 genes downstream of ORF24 of each contig, if present. This yielded a total of 1784 protein sequences, which were clustered at 15% sequence similarity using psi-cd-hit with the parameter -c 0.15 [18]. Protein clusters, which recruited proteins from at least 37 plasmids (>90% of the 41 apHPVs), were considered “core” genes. Clusters that recruited from between 50% and 90% of plasmids (at least 21) were considered “shell” genomes, and clusters that recruited from between 10% and 50% (at least 5) were considered cloud“ genomes. The core, shell, and cloud genome form, together, the pangenome. Protein alignments to calculate the pairwise amino acid identity of apHPV homologs of core proteins were computed using MUSCLE [19] with default parameters.

### 2.5. Phylogenetic Analysis

A distance matrix of the 41 apHPVs was calculated using the presence and absence of the pangenome using the ‘hclust’ function with the ‘average’ method in R. Based on this matrix, a phylogenetic tree was constructed using the ‘ggtree’ R library [20]. A second tree was calculated based on the host genomes of the plasmids. This was conducted by calculating an all genome alignment using the ‘gtdbtk classify_wf’ command, part of the GTDB-Tk v2 suite [21] with standard parameters, which was converted into a distance matrix using the IQ-TREE webserver [22] with the parameters ‘-bb 1000 -alrt 1000′. The tree was visualized using ‘ggtree’.

### 2.6. Gene Sharing Network of apHPVs, Archaeal Viruses, and Archaeal Plasmids

First, the following search query was used in order to retrieve archaeal plasmids from NCBI: ‘“archaeal”[Organism] OR “Archaea”[Organism]) AND “plasmid”[Filter] NOT “shotgun”[All Fields] NOT “MAG”[All Fields]’ (date of retrieval: 22 September 2023). Plasmids between 20 kbp and 70 kbp were selected for detailed analysis. This yielded 134 archaeal plasmids, which were used as the input for vConTACT2 [23], together with all complete apHPVs. Parameters were set to ‘--rel-mode ‘Diamond’ --db ‘ArchaeaViralRefSeq207-Merged’ --pcs-mode MCL --vcs-mode ClusterONE’. The network was visualized with flourish studio (https://flourish.studio/). Sequences in the cluster that contained both viral and plasmid-like elements were analyzed using VirSorter2 [24] with default parameters. Sequences that scored > 0.9 and showed more than three viral hallmark genes were considered viral.

### 2.7. Functional Annotation of apHPV Genes

Open reading frames were predicted using Prodigal (https://github.com/hyattpd/Prodigal, accessed on 14 August 2023 [25]) using the metagenomic flag -p meta. Functional annotation was conducted using DRAM (https://github.com/WrightonLabCSU/DRAM, accessed on 16 August 2023 [26]) with the following parameters: *–prodigal-mode meta –min-contig-size 10,000*. Protein sequences were searched against the NCBI nr database of non-redundant proteins using diamond blastp (https://github.com/bbuchfink/diamond [13]) and the following parameters: *--max-target-seqs 3 --evalue 0.00001 --fast.* Lastly, proteins were also searched against a set of databases (CDD, COILS, Gene3D, HAMAP, MOBIDB, PANTHER, Pfam, PIRSF, PRINTS, PROSITE, SFLD, SMART, SUPERFAMILY, and NCBIFAM) using InterProScan [27] with default parameters.

In addition to the aforementioned annotation steps, the protein structures of seven representatives of ORF8, ORF9, ORF17, ORF21, ORF23, and ORF24 were predicted and searched against protein structure databases using the Foldseek web server [28] (Appendix A). Since the web server only allows for proteins shorter than 400 amino acids (aa), in cases where the protein was longer than 400 aa, it was manually split into an N-terminal and an C-terminal part, each consisting of 400 aa. Protein structures were searched in 3Di/AA mode. For each protein, the two best hits (ranked by a low evalue and high Foldseek probability value) with an available annotation were selected, preferably from the AFDB50 database. Structures of ORF7 and ORF25, and their homologs, were downloaded from the AlphaFold protein structure database (https://alphafold.ebi.ac.uk/, accessed on 6 September 2023). For ORF6 homologs, the protein structures were predicted using AlphaFold 2 [29] as protein monomers. Protein structures were visualized with the open-source PyMOL(TM) Molecular Graphics System, Version 2.2.0 (L. DW. The PyMOL molecular graphics system). Structural predictions were manually inspected, and low-confidence regions were removed (pLDDT < 50). Subsequently, selected models were used for structural similarity comparison with DALI [30].

### 2.8. Identification of Clusters of Orthologous Groups (COGs)

The functional potential of two sets of genes was compared against each other: Set 1 was comprised of apHPV genes outside of the core region (from ORF6 to ORF25), and set 2 contained genes from 993 whole haloarchaeal genomes, retrieved from NCBI (NCBI genome site, filtered for taxon ‘183963’, *Halobacteria*). Genes were compared by BLASTp against the COG database (version 2020, accessed on 10 January 2023) [31] using diamond blastp with an evalue cutoff of 10^−5^. Each gene was assigned to one or more COG categories, resulting in functional profiles for the two sets of genes. For each category, a relative abundance was calculated as follows:$${\mathrm{rel}(\mathrm{set})}_{\mathrm{cat}}=\frac{n_{\mathrm{hc}}}{n_{\mathrm{tot}(\mathrm{set})}},$$
where $n_{c}$ is the number of hits per category and $n_{\mathrm{tot}(\mathrm{set})}$ is the total number of hits for this category. Then, the difference in relative abundances was calculated per category:
$$\Delta {\mathrm{rel}}_{\mathrm{cat}}=\mathrm{rel}(\mathrm{apHPV})-\mathrm{rel}(\mathrm{halo}).$$

### 2.9. Detection of Antiviral Defense Systems and Anti-CRISPR Proteins in apHPVs

Antiviral defense systems were identified using the PADLOC webserver with the corresponding PADLOC database (v2.0.0) [32]. Detected defense systems for each apHPV are listed in Appendix A. Anti-CRISPR proteins (arcs) were detected by diamond blastp searches of the apHPV proteins against the Anti-CRISPRdb (version 2.2) [33]. Additionally, the core regions (between ORF6 and ORF25) of all apHPVs were searched for matches against the IPhoP CRISPR-spacer database with blastn with default parameters. Hits with >97% identity and length > 30 were considered to be positives.

### 2.10. Number of Plasmids per Class

In order to calculate the number of plasmids per haloarchaeal class, NCBI was searched with the following search-query: “archaeal”[Organism] OR “Archaea”[Organism] AND “plasmid”[Filter] NOT “shotgun”[All Fields] NOT “MAG”[All Fields]”. This yielded 1134 plasmids, from which the taxonomy was derived. The number of genomes on GTDB for each class was retrieved through the built-in tree-browser (https://gtdb.ecogenomic.org/tree?r=p__Halobacteriota). On the day of retrieval (22 September 2023), the number of genomes was: *Archaeoglobi* (71), *Halobacteria* (750), *Methanobacteria* (559), *Methanococci* (61), *Methanomicrobia* (311), *Thermococci* (141), *Thermoplsamata* (801), and *Thermoprotei* (122).

## 3. Results

### 3.1. Selection of Core Proteins of pR1SE

In order to detect pR1SE-like elements in public databases and in our own dataset of metagenomes from Australian salt lakes, HMM profiles of six proteins of the original pR1SE were created. The proteins selected for this search—ORF6, ORF8, ORF17, ORF21, ORF23, and ORF24—were detected in pR1SE-containing vesicles (PVs) by mass spectrometry and were suggested to be structural proteins of PVs by Erdmann et al. [4]. For each of the six ORFs, a homolog was detected in *Halorubrum saccharovorum*, *Haloterrigena turkmenica*, and *Halopiger xanaduensis*, resulting in a protein set that was used to create the respective HMM profiles as described in Methods Section 2.2.

### 3.2. Identification of Forty Complete pR1SE-like Mobile Genetic Elements Using an HMM-Based Approach

Using the HMM profiles of the core proteins, 40 complete (at least five proteins present) pR1SE-like entities were detected. Additionally, 38 ‘incomplete’ pR1SE-like elements were detected, with a minimum of 3 of the 6 proteins detected (Appendix A). Of the 40 complete pR1SE-like entities, 35 were retrieved from NCBI, while five additional contigs were detected in the metagenomes of Australian salt lakes [10], and none could be detected in the IMG/VR database. We will refer to the detected pR1SE-like entities as apHPVs, for archaeal plasmids of haloarchaea potentially transferred in plasmid vesicles, while ‘H’ can be replaced by the abbreviation for the host (e.g., HR for *Halorubrum*; apHRPV1 for pR1SE) as common for virus names. The 40 apHPVs originate in hypersaline environments all around the world, e.g., salt lakes, salt mines, salted food, or marine solar salterns. More than half (26) were found as part of plasmids with a size below 100,000 bp, twelve on plasmids between 100,000 bp and 500,000 bp, two were integrated into secondary chromosomes (766 kbp and 596 kbp, respectively), and one was integrated into a main chromosome (3.7 mbp). For 23 out of 41 apHPVs, the respective contigs were circular, either by NCBI annotation or by our analysis (see Methods Section 2.3). The average GC content ranged from 57.6% to 64.8%. The apHPVs and incomplete relatives are listed in Appendix A.

### 3.3. apHPVs Exhibit Two Highly Conserved Gene Clusters Interspersed by a Variable Region

All complete apHPVs are organized into two gene clusters (Figure 1). Both gene clusters contained proteins used for the search and additional ORFs conserved across the majority of apHPVs that we define as core proteins of apHPVs. ORF numbers refer to the original pR1SE annotation (accession KX687704.1). Gene cluster 1 contains ORF6–9, and gene cluster 2 contains ORF17, ORF21, and ORF23–25. The synteny of genes within a cluster and the order of the two gene clusters were conserved in all identified complete apHPVs. However, the region between the two clusters (between ORF9 and ORF17) was highly diverse, consisting of three to eleven genes that were rarely shared between different apHPVs (Appendix A). Interestingly, all of the 38 incomplete pR1SE-like elements detected in our search contained either genes from cluster 1 (34) or genes from cluster 2 (3), with only one contig containing ORFs 6 to 8 and ORF17 (Gairdner_NODE_4298_length_11003_cov_3.762240).

### 3.4. Phylogeny and Host Association of apHPVs

All apHPVs were identified in species belonging to the class *Halobacteria,* of the order *Natrialbales* (n = 18), *Haloferacales* (n = 9), and *Halobacteriales* (n = 11). Detailed taxonomic information is given in Appendix A, and family rank is indicated in Figure 2. Host information was retrieved directly from NCBI if possible. For metagenome-derived sequences, for which host information was not available, hosts were predicted using IPhoP. For three of the five metagenomic sequences, a host could not be predicted (Appendix A). A cluster analysis of the apHPVs was calculated based on the presence and absence of genes in the cloud genome (present in more than 10% of all plasmids) of each apHPV (Figure 2). Interestingly, we found two examples of two different apHPVs co-existing in the same host: *Halorubrum saccharovorum* DSM 1137 and *Halorubellus salinus* GX3 (Appendix A).

### 3.5. Replication of apHPVs

The plasmid pR1SE contains two proteins that might be responsible for plasmid replication. ORF1 encodes a predicted RepH plasmid replication protein, and ORF29 is an CDC6/ORC1-like protein that is commonly involved in the initiation of replication of haloarchaeal chromosomes and plasmids [34]. We searched the newly discovered apHPVs for proteins involved in replication. For 33 of 41 plasmids, a potential replication protein was found close (+/−10 ORFs up- or downstream) to the core region (Figure 2). Most commonly, proteins belonging to the minichromosome maintenance complex (MCM) were identified (n = 20). A CDC6/ORC1-like protein was detected on ten apHPVs. In two cases, a HNH endonuclease and a recombinase were found close to the core region. For seven apHPVs, no obvious candidate for a replication protein was detected. Potential origins of replication (direct repeats, interspersed direct repeats, or palindromic repeats of at least 25 bp) were detected in 16 plasmids, either directly before ORF6 or just after ORF25. None of the newly identified apHPVs exhibits a homolog to ORF1 of pR1SE. About ~100 aa (aa 340–440) of the predicted tertiary structure of ORF1 shows good alignments with DNA-binding proteins (transcriptional regulators, helicase, chromosome partition protein, CDC6/Orc1-like), indicating that it could indeed be DNA-binding and involved in plasmid replication.

### 3.6. Core Genes and Their Predicted Functions

A total of ten genes were conserved in nearly all (>90%) plasmids, which were therefore considered the core genome of apHPVs (Table 1 and Figure 1). All of the protein products of the core genes, except one (ORF25), were detected as potentially structural proteins in PVs. However, their role in the generation of PVs remains unknown. ORF7 was consistently (amongst all apHPVs) annotated as a S8 family serine peptidase based on conserved domains, as reported previously [4]. However, all other ORFs showed no similarity to known proteins on the amino acid sequence level or very inconsistent results across homologs of different apHPVs. Therefore, we used the predicted tertiary structures [29] and searched for related structures in public databases [25,27] (see Method Section 2.7). This approach confirmed the annotation of ORF7 as a subtilisin-like serine protease, because the structures of ORF7 and apHPV homologs consistently hit subtilisin-like serine protease homologs. ORF8 and its homologs in other apHPVs are best aligned (~aa 1–180, see Appendix A) with the structure of Necrosis and Ethylene-inducing Protein 1 (NEP1). Predicted structures of ORF9 homologs were very poorly resolved (pair-wise AAI of ~18%), leading to inconsistent hits; however, the best hits included S-layer proteins (~aa 30–200) and spore coat protein CotH (IPR014867). The N-terminus of ORF23 (aa 50–150) and its homologs consists of eight beta strands forming two putative beta sheets, and hits putative pilus assembly proteins (Type II and Type III secretion systems) and TonB receptor proteins. The predicted structures of ORF25 and its homologs align (aa 135–565) (RMSD values between 4–5) with components of the Type VI secretion system (VirB4) and the bacterial conjugation protein TRWB, but also with HerA (RMSD 5.0) [35] and the genome-packaging NTPase of Sulfolobus turret icosahedral virus 2 [36] (RMSD 2.7). Different apHPV homologs of ORF17 yielded inconsistent hits. The best alignments were against Ricin B-type lectin, PKD, SH3, and UNBV_ASPIC domain-containing proteins. For ORF21 and ORF24, no homologs, neither on a sequence nor on a structural level, were found.

ORF6, the largest conserved protein, was previously proposed to be one of the major structural proteins in PVs. Surprisingly, ORF6 proteins presented a relatively low sequence similarity (pairwise AAI ~18.3%) and high variability. While ORF6 of pR1SE has a WD40 domain (IPR001680) at the N-terminus (Appendix A), this domain was present in only ~56% of the predicted structures of ORF6 homologs. Meanwhile, the other fraction presented variable N-terminal domains, often in the form of extended arrays of antiparallel beta sheets (Appendix A). On the other hand, the C-terminal domain of ORF6 homologs was highly conserved, composed of multiple arrays of antiparallel beta sheets facing each other (Appendix A) that likely form the hydrophobic core of the protein. Homology searches against protein structure databases using Foldseek and DALI revealed this array to be unique to ORF6-like proteins, as no matches with similar folds to the whole array were obtained. Significant matches to smaller subsections of the C-terminal domain include a chromosome condensation regulator RCC1 (A0A2Z4FI68; score 141) and a pair of lipoproteins of bacterial origin (UniProt F3ZI24 and A0A1H8ZS84; score 130 and 128 respectively). Despite the high degree of structural conservation among the C-termini of ORF6 homologs, their function remains unknown.

### 3.7. Transfer of Host Genetic Material by apHPVs

Plasmid pR1SE was shown to integrate into host chromosomes or other replicons. Upon excision, it often included host genomic material, resulting in pR1SE derivatives with variable extensions [4]. Three of the novel apHPVs were also found integrated into long replicons (>500 kbp), while the remaining 38 entities were found on plasmids ranging from 44 kbp up to 271 kbp. The core region (gene clusters 1 and 2, and the flexible region in between) of apHPVs range from a minimum of 18 kbp to 44 kbp (Figure 1 and Appendix A). The plasmids carrying the core region are up to eight times larger than the respective core region. We analyzed the additional genetic material and found that it often matched host sequences with up to 100% identity. Comparing the functional profiles of transported (non-core) regions of apHPVs in comparison to haloarchaeal genomes revealed that apHPVs contain significantly higher amounts of genes associated with replication, recombination and repair, defense mechanisms, the mobilome (including transposases and viral-like genes), transcription, cell motility, cell cycle, and control (Figure 3). In contrast, genes related to metabolism and translation are present much less frequently in apHPV-associated genomic information.

### 3.8. Antiviral Defense Systems and Anti-CRISPR Proteins on apHPVs

In sixteen of the 38 complete, non-integrated apHPVs, a putative antiviral defense system was identified. The defense systems were located adjacent or very close (+/− 10 ORFs up- or downstream) to the core region (Appendix A). Among the systems detected were CRISPR-Cas systems [37], cyclic GMP-AMP type systems [38,39], a complete phosphorothioation modification module [40], and other defense systems (e.g., restriction modification systems) [41,42]. Only one apHPV (AOID01000019) contained an anti-CRISPR protein (AcrIIA7, WP_081009421.1), notably within the core region of that apHPV. Conversely, searching the core-regions for CRISPR-spacer hits revealed that seven out of 41 apHPVs are targeted by CRISPR-systems in haloarchaea. Most often (6 out of 7) apHPVs were targeted by a close taxonomic relative of their host organism, and in one case, the host was targeting its own intracellular apHPV.

### 3.9. Clustering of apHPVs with Archaeal Plasmids and Archaeal Viruses

While the pR1SE plasmid showed a virus-like lifestyle, being transferred in virus-like particles, no virus-like genes, e.g., genes encoding for virus capsid proteins, could be identified on the plasmid, not even through analysis of predicted tertiary structures. To determine the positioning of apHPVs between viruses and plasmids, we generated a gene-sharing network between archaeal plasmids, archaeal viruses, and the discovered apHPVs (Figure 4). This resulted in clusters composed of either only viruses, only plasmids, or only apHPVs, in all but one case. One cluster containing archaeal virus BJ1 and *Halorubrum* phage CGphi46, among others, also contained sequences labeled as plasmids in public databases. However, we found these sequences to be of viral origin using viral prediction tools (see method Section 2.6, score > 0.9 with at least 3 viral hallmark genes). apHPVs clustered into a single large cluster, with connections to both a cluster of haloarchaeal viruses and a cluster of haloarchaeal plasmids. The genes shared between apHPVs and the other sequences were found to be outside the core region.

## 4. Discussion

Upon the discovery of pR1SE, only one other organism with similar proteins could be detected in public databases [4]. Our database search using HMM profiles of six pR1SE proteins revealed at least 41 full-length apHPVs (including pR1SE) and a number of contigs that represent partial apHPVs, indicating that pR1SE is more widespread than a single case of a non-conjugative, vesicle-forming plasmid. The apHPVs seem to be restricted to archaea belonging to the class of *Halobacteriales* and were found in hypersaline environments all over the world.

The genetic organization of the ‘core region’ in two syntenic gene clusters is conserved among all complete apHPVs. The core region contains all proteins that were detected as potential structural proteins in PVs, indicating that other apHPVs could also be capable of PV formation. However, whether all of the detected plasmids indeed disseminate via self-encoded plasmid vesicles will require experimental verification. We also identified a significant number of apHPV-like entities that contain only one of the clusters, with the majority containing the first cluster (ORF6–ORF9). These could be remnants of inactive integrated apHPVs. Similar to integrated defective proviruses, they likely have an advantage for the host; otherwise, the regions would not be maintained.

The region between the two conserved gene clusters of the core is very flexible and differs between apHPVs, not only in size but also in gene content. Both virus and plasmid genomes are often organized in modules that have different functions [43,44]. Plasmid genomes usually have modules that are responsible for plasmid replication and maintenance and modules that are required for plasmid transfer. Both gene clusters in the core region contain proteins that were identified as structural proteins in PVs by mass spectrometry; therefore, we previously suggested that both clusters are involved in PV formation and plasmid dissemination [4]. However, the clear organization into two clusters, interspersed by a variable region, might indicate that the two clusters have distinct functions.

A conserved module for plasmid replication could not be identified on apHPVs. Instead, the majority of the apHPVs encode for either a CDC6/ORC1 or a MCM helicase, which have most likely been acquired from a host genome. Haloarchaea are known to exhibit multiple origins of replication and of CDC6/ORC1 proteins within one genome, and those can likely be transferred between different haloarchaea [34]. Direct repeats of ~25 bp in proximity to the core region were detected in nearly all apHPVs and could serve as the ORI of each plasmid, possibly allowing replication even if no plasmid-encoded CDC6/ORC1 is present. Replication of haloarchaeal megaplasmids or secondary chromosomes is driven by CDC6/ORC1 and associated ORIs, and they rarely reach copy numbers that are significantly higher than those of the main chromosome (up to 2-fold) [45,46]. Plasmid pR1SE was shown to reach up to 15 copies per host chromosome copy [7]. It remains to be elucidated whether this is due to ORF1 activity or whether other apHPVs without ORF1 can also reach higher copy numbers.

The functional prediction of the identified core proteins remained challenging, despite the advancement of tools for tertiary structure prediction and structural comparison. However, structural analysis revealed conserved domains in some of the proteins. ORF6 was thought to be a central structural and potentially coat-forming protein in PVs, recruiting other structural proteins and the cargo. The prediction was based on the N-terminal WD40 domain, which is also present in coat proteins of eukaryotic intracellular vesicles [47]. However, our analysis of ORF6 in other apHPVs revealed that the N-terminus is highly diverse, and only half of the other homologs contain a WD40 domain. In contrast, the structure of the C-terminus of the protein appears to be highly conserved, but no structural homologs could be identified in public databases. Thus, the role of ORF6 in PV formation remains unresolved. ORF7, for which the annotation as a subtilisin-like serine protease was supported by structural predictions, was detected in all size fractions of PVs separated on polyacrylamide gels [4], indicating a strong interaction with other PV proteins. Therefore, we suggest that ORF7 could potentially be involved in the maturation of different pR1SE proteins, resembling viral proteases involved in maturation [48]. The structure of ORF8 was conserved amongst different apHPVs and showed good alignments with the structure of NEP1-like proteins (Necrosis and ethylene-inducing proteins), or NLPs. NLPs proteins contain NPP1 domains (necrosis-inducing Phytophthora protein), are secreted by phytopathogenic bacteria and fungi, and contribute to host infection by plasma membrane permeabilization [49]. Proteins with NPP1 domains have been identified in the extracellular vesicles of phytopathogenic fungi [50]. While the detailed molecular function of NLPs for the producing organisms is still unknown, it is known that NLPs interact with the plant membrane and are involved in membrane permeabilization. We, therefore, assume that ORF8 is membrane-interacting and may even have membrane-remodeling functions. ORF9 was predicted to build a stabilizing outer coat of PVs, and our reanalysis did not reveal any other potential function. The predicted structures of ORF17, ORF21, and ORF24 homologs were inconsistent and did not allow the identification of structural homologs in public databases, thus prohibiting a reasonable prediction of function. The N-terminus of ORF23 is highly conserved and shows similarity to domains within pilus assembly proteins and TonB receptor proteins. However, the function of ORF23 still remains elusive because the function of this particular domain is unknown. The structure of ORF25 is highly conserved across all apHPVs and is similar to both a viral DNA-packaging ATPase and ATPases involved in plasmid recruitment during conjugation. ORF25 was not detected in PVs; therefore, we predict that ORF25 is responsible for the recruitment of apHPVs into PVs. Both ORF23 and ORF25 showed similarities to components of Type II or Type IV secretion systems, respectively, which could indicate that they are acting together to either secrete or pull the cargo into PVs. Considering the genomic organization into two clusters (ORF6–9 and ORF17, ORF21–ORF25) and the fact that proteins of cluster 1 and ORF17 were more abundant in PVs [4], we suggest that cluster 1 could play a major role in the formation and structure of PVs, while cluster 2 could be responsible for plasmid packaging. ORF17 was highly diverse amongst different apHPVs and could possibly be responsible for host cell recognition.

While we have no experimental evidence that apHPVs other than pR1SE are transferred via PVs, it is evident from our data that other apHPVs also integrate into host chromosomes or secondary replicons because we found examples of integration. Additionally, the size of the majority of apHPVs exceeds significantly the size of the core region, indicating that they have recruited a substantial amount of genomic information from earlier integration events (Figure 5B). Integrases were detected on most apHPVs; however, these integrases could have also been retrieved from the host, as in the case of pR1SE, without being responsible for the active integration of apHPVs. Integration, excision, and uptake of host genomic information were shown for pR1SE [4], and we infer that other apHPVs exhibit the same capability (Figure 5A). Considering the size of the newly discovered apHPVs, we suggest that apHPVs are responsible for the horizontal transfer of very large fragments of genomic DNA. Notably, this function coincides with the high general genomic flexibility of haloarchaea and the significantly higher number of plasmids per genome in this class (Figure 5C). This could indicate that elements like apHPVs are either responsible for the high genomic flexibility or that apHPVs could only evolve because recombination events were already happening at an increased rate.

The genomic composition of the non-core regions of apHPVs is similar to the flexible regions of the host genome [8,51], enriched in genes associated with replication, recombination and repair, defense mechanisms, motility, transcription, and the general mobilome. Additionally, we found virus defense systems on almost all apHPVs. Therefore, we suggest that apHPVs could play an important role in the transfer of virus defense systems between cells, and hosts could strongly benefit from apHPVs, which support their defense against viruses. While we did find CRISPR spacers against the core region of apHPVs, only a few were targeted, indicating that apHPVs are either often not recognized as invading genetic elements or interfere with the uptake of new spacers by the CRISPR system. One apHPV was found in host organisms encoding a spacer against the same apHPV, and one anti-CRISPR protein was identified, suggesting that apHPVs also counteract CRISPR interference. We found two cases of two different apHPVs co-existing in one host, demonstrating that they do not exclude each other as is known for some viruses (superinfection exclusion).

Strikingly, the gene sharing network of archaeal plasmids, viruses, and apHPVs revealed that apHPVs could potentially exchange genetic material not only with host genomes and plasmids but also with viruses. However, it remains to be elucidated whether apHPVs pick up regions of integrated viruses or recombine with actively replicating viruses. Considering that the life cycle of apHPVs already places them at the interface between viruses and plasmids, their ability to recombine with both plasmids and viruses further strengthens the emerging connection between viruses and plasmids that have long been regarded as very distinct and unrelated MGEs.

## 5. Conclusions

In our study, we identified 40 pR1SE-related elements, apHPVs, in haloarchaea, showing that this class of elements is more widespread than previously thought. All apHPVs share a very conserved genetic organization, including the majority of proteins that were detected in PVs of pR1SE. It is thus conceivable that all apHPVs could potentially be disseminated in PVs. However, even modeling the tertiary structures of these core proteins often did not allow us to draw conclusions about their function, pointing towards a unique mechanism. The potential of pR1SE for horizontal gene transfer was already demonstrated by Erdmann et al. [4], and the analysis of the additional apHPVs detected in this study revealed even more complex interactions. The apHPVs appear to exchange genetic material not only with the host but also with archaeal plasmids and viruses, possibly playing an important role in the evolution of both plasmids and viruses.

## Figures and Tables

**Figure 1 microorganisms-12-00005-f001:**
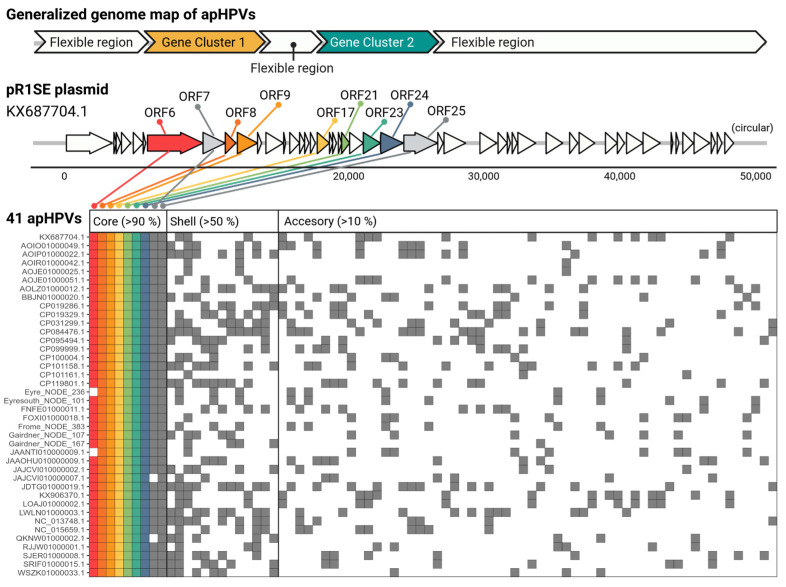
Generalized genome map and cluster analysis of proteins belonging to the core, shell, and cloud genomes of apHPVs. On top, a generalized genome of apHPV is shown, with a flexible region between two conserved gene clusters and flexible regions surrounding this core region. Below, the genome of the original pR1SE (KX687704.1) is drawn, where the core proteins, identified in >90% of all apHPVs, are highlighted. Below a presence-absence plot of protein clusters (columns) in different apHPVs (rows) is presented. Note that the homologs of ORF6, ORF8, ORF9, ORF21, and ORF24 did not cluster at the selected threshold (15% AAI), but were split into multiple clusters and collapsed for better visualization. An expanded view of the core proteins is given in Appendix A.

**Figure 2 microorganisms-12-00005-f002:**
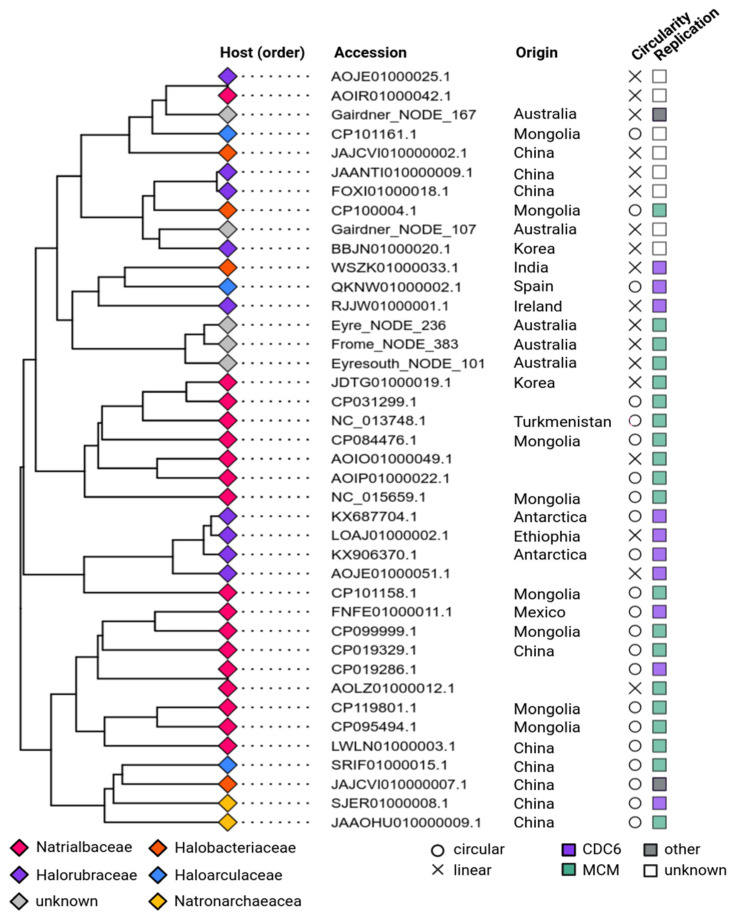
Cluster analysis and additional features of complete apHPVs. The tips of the branches are labeled according to the host associated with the plasmid (taxonomic rank: family). The country of origin is indicated in the second column. Circularity and the identified putative replication gene are shown in the columns to the right. CDC6—CDC6/ORC1-like protein; MCM—Minichromosome maintenance complex.

**Figure 3 microorganisms-12-00005-f003:**
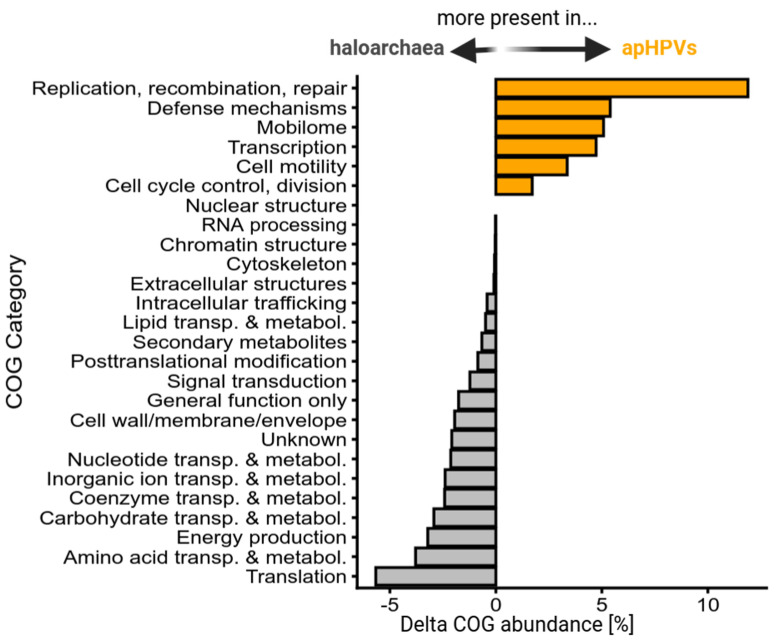
Functional profile of apHPV-transported genes in comparison to host genes. The difference between the percentages of genes associated with one COG category is for two sets of genes: Genes from 993 halobacterial genomes and genes from 38 apHPV non-core regions, excluding 3 apHPVs that were integrated into larger replicons (see Methods Section 2.8). Positive values indicate an increased relative abundance of this COG category in apHPVs, while negative values indicate an increased relative abundance of this COG category in haloarchaeal genomes.

**Figure 4 microorganisms-12-00005-f004:**
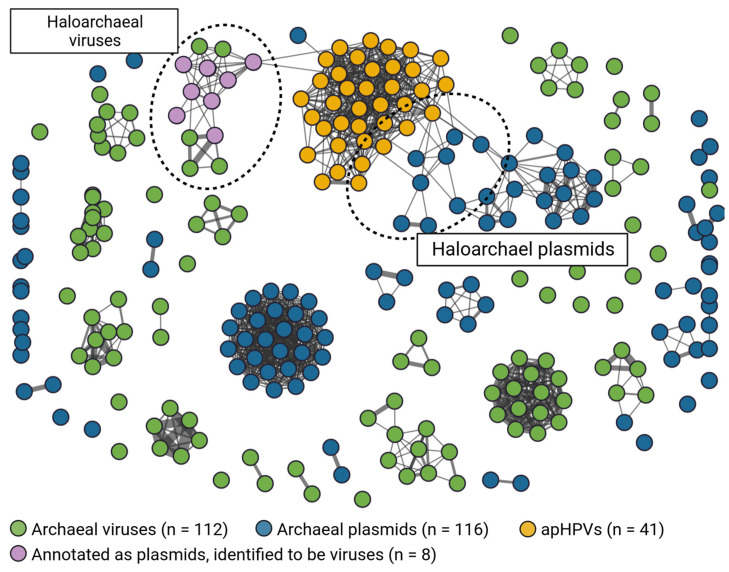
Gene sharing network between archaeal plasmids, archaeal viruses, and apHPVs. The mixed cluster of viruses and plasmids (top left) was found to be exclusively viral, with some virus genomes mistakenly classified as plasmids. apHPVs share non-core genes with both a viral cluster and haloarchaeal plasmids.

**Figure 5 microorganisms-12-00005-f005:**
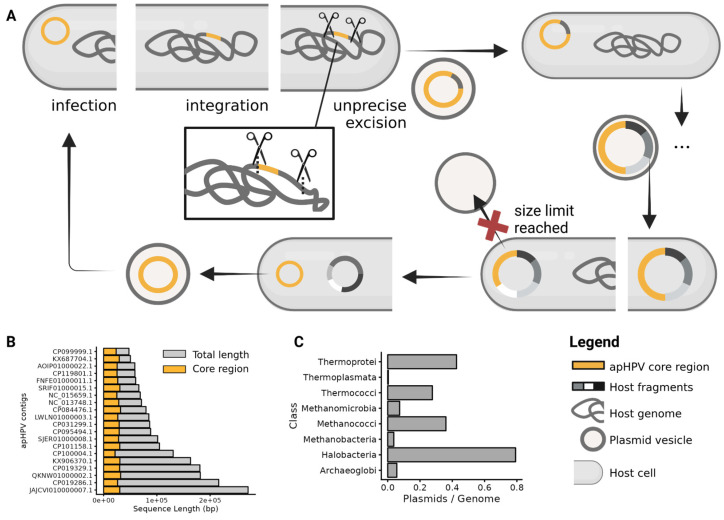
The proposed replication cycle of apHPVs is based on the replication cycle of pR1SE [4]. (**A**)—Clockwise replication cycle of apHPVs: A vesicle containing an apHPV plasmid infects a plasmid-free host. The plasmid integrates into the main chromosome, secondary replicons, or replicates independently. During excision out of host replicons, genomic fragments of the host genome are co-excised and incorporated into the plasmid. This infection-integration-excision cycle potentially occurs multiple times, leading to an increased size of the plasmid, which contains genetic material from multiple hosts. At some point, the plasmid becomes too large to be packaged into PVs (red cross), only the core region of apHPVs excises, and the additional genetic material remains within the host. If an ORI is present, the remains could replicate as secondary replicons. (**B**)—Length (base pairs) of the core region and the total length of apHPVs. Only circular, non-integrated apHPVs are shown (n = 20). (**C**)—Number of plasmids on NCBI per archaeal class; divided by the number of genomes of this class on GTDB.

**Table 1 microorganisms-12-00005-t001:** Core genes of apHPVs. A detailed list of structural homologs is given in Appendix A.

ORF No. ^1^	Pairwise AAI ^2^	Best Hit	Method of Annotation
ORF6	18.3%	-	AlphaFold/DALI
ORF7	29.1%	subtilisin-like serine protease	foldseek/AlphaFold/DALI
ORF8	23.8%	Necrosis and ethylene inducing protein 1	foldseek
ORF9	17.5%	S-layer/Big6/CotH	foldseek
ORF17	34.2%	PKD/SH3/Ricin B-type lectin/UnbV_ASPIC	foldseek
ORF21	26%	-	foldseek
ORF23	31.5%	pilus assembly proteins/TonB receptor proteins	foldseek
ORF24	23.9%	-	foldseek
ORF25	41%	VirB4/helicase/genome packaging ATPase	foldseek/AlphaFold/DALI

^1^ According to the original annotation of pR1SE (accession KX687704.1) and homologs in other apHPVs. ^2^ Based on a protein alignment of all homologs of a respective ORF in detected complete apHPVs.

## Data Availability

All scripts, plots, intermediate results, and pipelines are deposited on github: https://github.com/dluecking/pR1SE_australian_lakes.

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
