# Peer review of "Distribution and Implications of Haloarchaeal Plasmids Disseminated in Self-Encoded Plasmid Vesicles"

_microorganisms, 2023, doi:10.3390/microorganisms12010005_

Round 1

Reviewer 1 Report

Comments and Suggestions for Authors

The article contains novel data on MGEs, which in turn is intresting from the point of view of both halophilic archaea microbiology and archaea genetic and genomic in general. As the maniscript does not contain any critical gaps, it may be accepted without serious revisions. 

Some points may be commented:

1. Line 223. “The proteins selected for this search - ORF6, ORF8, ORF17, ORF21, ORF23, ORF24 - were detected in pR1SE-containing vesicles (PVs) and were suggested to be structural proteins of PVs.” – Could you briefly explain this point?

2. Line 418 “The apHPVs seem to be restricted to archaea belonging to the class of Halobacteriales and were found in hypersaline environments all over the world.” Have similar MGEs been found in any other archaea/bacteria?

Comments on the Quality of English Language

The manuscript contains some typos.

Author Response

Reply is attached as pdf file

Reviewer 2 Report

Comments and Suggestions for Authors

The authors found multiple elements similar to the plasmid pR1SE in the high-salt environment of the world, which provides possible evidence to support the theory that plasmids and viruses are not distinct, but closely related, and mobile genetic elements. However, present version is not allowed for considering publication. Several uncertain information needs to be clarified. The manuscript might be published to journal if the authors would be able to modify manuscript in accordance with several comments below.

1) There are too many formatting problems in the text, such as repeated punctuation, etc.

2)Tell us a little bit about what is a hidden Markov model (HMM).

3) Why are sediment crusts from highly salinity lakes sampled?

4) In Fig. 3, the authors said that they generated a gene sharing network between archaeal plasmids, archaeal viruses and the discovered apHPVs. How did they generated a gene sharing network, and how does it working?

Other questions:

>1. What is the main question addressed by the research?

A: The biggest issue with this manuscript is its lack of innovation

>2. Do you consider the topic original or relevant in the field? Does it address a specific gap in the field?

A:This research is relevant to this field and to some extent reveals the distribution of halogenated archaeal plasmids in self encoded plasmid vesicles

>3. What does it add to the subject area compared with other published material?

A:This manuscript provides a more precise explanation of the role of apHPV in the evolution of viruses and plasmids in halogenated archaea, further supporting that plasmids and viruses are not distinct but closely related mobile genetic elements.

4. What specific improvements should the authors consider regarding the methodology? What further controls should be considered?

A:The author should delve into the close relationship between plasmids and viruses as mobile genetic elements.

>5. Are the conclusions consistent with the evidence and arguments presented and do they address the main question posed?

A:The conclusion is basically consistent with the evidence and arguments presented, solving the main issues raised, but can provide more discussion

>6. Are the references appropriate?

A:Inconsistent citation format of reference

>7. Please include any additional comments on the tables and figures.

A: No comments.

Comments on the Quality of English Language

Minor editing of English language required

Author Response

Reply is attached as pdf file
